# Discovery of an ene-reductase for initiating flavone and flavonol catabolism in gut bacteria

Gaohua Yang[1,2,4], Sen Hong [2,3,4], Pengjie Yang[1,2,4], Yuwei Sun[1], Yong Wang[1], Peng Zhang[3], Weihong Jiang [1✉] & Yang Gu [1✉]

Gut microbial transformations of flavonoids, an enormous class of polyphenolic compounds abundant in plant-based diets, are closely associated with human health. However, the enzymes that initiate the gut microbial metabolism of flavones and flavonols, the two most abundant groups of flavonoids, as well as their underlying molecular mechanisms of action remain unclear. Here, we discovered a flavone reductase (FLR) from the gut bacterium, *Flavonifractor plautii* ATCC 49531 (originally assigned as *Clostridium orbiscindens* DSM 6740), which specifically catalyses the hydrogenation of the C2–C3 double bond of flavones/flavonols and initiates their metabolism as a key step. Crystal structure analysis revealed the molecular basis for the distinct catalytic property of FLR. Notably, FLR and its widespread homologues represent a class of ene-reductases that has not been previously identified. Genetic and biochemical analyses further indicated the importance of FLR in gut microbial consumption of dietary and medicinal flavonoids, providing broader insight into gut microbial xenobiotic transformations and possible guidance for personalized nutrition and medicine.

[1] CAS-Key Laboratory of Synthetic Biology, CAS Center for Excellence in Molecular Plant Sciences, Shanghai Institute of Plant Physiology and Ecology, Chinese Academy of Sciences, Shanghai 200032, China. [2] University of Chinese Academy of Sciences, Beijing, China. [3] National Key Laboratory of Plant Molecular Genetics, CAS Center for Excellence in Molecular Plant Sciences, Shanghai Institute of Plant Physiology and Ecology, Chinese Academy of Sciences, Shanghai 200032, China. [4] These authors contributed equally: Gaohua Yang, Sen Hong, Pengjie Yang. ✉email: wjiang@cemps.ac.cn; ygu@cemps.ac.cn

The human gut microbiota has evolved to effectively transform various dietary foods and xenobiotics including pharmaceuticals and environmental chemicals, and have therefore been extensively connected with human physiology and health[1,2]. Over the last few decades, studies have mainly focused on the associations between gut microbial xenobiotic transformations and host biology[3–5]. Despite the advances in our knowledge of the physiological importance of these microbe–host interactions, relatively little is known about the specific microbial strains, genes, and enzymes responsible for these activities. Fully understanding the molecular mechanism involved in transforming xenobiotics and their effects on host health and disease remains a great challenge.

Flavonoids are the most abundant polyphenols in plants[6]. According to their chemical structure, flavonoids may be roughly categorized into the following subclasses: flavones, flavanols, flavanones, flavonols, isoflavones, anthocyanins, and chalcones[7]. These compounds possess a great diversity of medicinal activities against diseases, such as cancer, inflammation, cardiovascular diseases[8–10], and the recently erupted novel coronavirus pneumonia[11]. Flavonoids, as dietary nutrients, also attract great interest due to their broad benefits to human health, such as antioxidant activity and improving immunity[12]. Studies of gut microbial xenobiotic modifications have revealed that these organisms contribute to the metabolism of flavonoids in the human body, thereby changing their bioavailability or generating metabolites that affect human health[13–15]. However, the underlying genetic and biochemical mechanisms remain minimally explored. The efforts in this aspect will require a full identification of genes and enzymes that mediate flavonoid metabolism in gut microbes.

Flavones and flavonols are the two main subclasses of flavonoids and have highly similar structures (Fig. 1a). Previous studies have identified a small number of gut bacteria, e.g., *Eubacterium ramulus* DSM 16296, which are capable of transforming these two compounds[16]. A recent work further revealed that the gut bacterium, *Clostridium orbiscindens* DSM 6740 (currently assigned as *Flavonifractor plautii* ATCC 49531), can generate desaminotyrosine (DAT) from the flavone apigenin to protect the human host from influenza[15], renewing our knowledge of the relevance of flavonoid compounds to human health. Despite broad interest, the enzymes involved in the gut microbial catabolism of these two compound classes are incompletely identified and understood[17], in which the enzyme sufficient for the initial step of the whole pathway, i.e., hydrogenation of the C2–C3 double bond on the C-ring (Fig. 1a) remains unknown. No existing reductase is reported to have the catalytic activity towards this reaction, and therefore, it is unclear whether this essential transformation step involves a distinct class of reductases.

Here, using a combination of bioinformatic, biochemical, and genetic analyses, we discovered and functionally characterized an unreported flavone reductase (FLR) that specifically catalyzes the hydrogenation reaction towards flavones and flavonols, and initiates their metabolism in *F. plautii* ATCC 49531 as a key step. The underlying molecular mechanism was intensively elucidated by analyzing the crystal structure of FLR as well as the co-crystal structures of FLR and its substrates. Of note, FLR and its widespread homologs represent an unreported class of ene-reductases with highly distinct amino acid sequence and catalytic property compared with those of all known ene-reductases. FLR-like enzymes were abundant in the human gut microbiota as well as other microbes living within flavonoid-rich environments, thereby suggesting a broad role of these ene-reductases in microbial communities.

## Results

**Discovery of the reductase (FLR) that initiates flavone catabolism in *F. plautii* ATCC 49531.** Considering the high structural similarity between isoflavones and flavones, we first hypothesized that the reported isoflavone reductase (IFR), which is capable of reducing the C2–C3 double bond of isoflavones (Supplementary Fig. 1a), also has catalytic activity towards flavones. Thus, we cloned a representative IFR enzyme, GmIFR, from soybean[18], and a putative IFR enzyme (sgrIFR, gene ID: SGR_2256) from *Streptomyces griseus*, and expressed these two enzymes in the *Escherichia coli* BL21 strain. The resulting *E. coli* transformants were cultivated in a defined medium containing apigenin (a main flavone) to examine whether they can degrade this compound. However, no detectable activities were observed for GmIFR and sgrIFR towards apigenin, while they could convert daidzein (a main isoflavone) to dihydrodaidzein (Supplementary Fig. 1b–e). These results strongly suggest that IFRs are incapable of hydrogenation of the flavone C2–C3 double bond and the enzyme initiating flavone catabolism may be highly specific and distinct from the known reductases involved in flavonoid metabolism. We next turned to bioinformatic analysis to search for this enzyme.

Chalcone isomerase (CHI) and phloretin hydrolase (PHY), two enzymes responsible for the downstream reaction steps in flavone metabolism (Fig. 1a), have been identified in *E. ramulus* DSM 16296[19,20]. We therefore screened all the proteins of the aforementioned apigenin-metabolizing *F. plautii* ATCC 49531 through the position-specific iterative (PSI)-BLAST using CHI and PHY from *E. ramulus* DSM 16296 as the input. The search results revealed two proteins (encoded by A4U99_05920 and A4U99_15225) homologous to the *E. ramulus* CHI and PHY, respectively (50.18% and 46.92% amino acid sequence identity, respectively).

Next, a further comparative genomic analysis using the webtool STRING revealed four and ten potential interacting partners (gene neighborhood, co-occurrence, and co-expression) to the *chi* and *phy* genes, respectively, in *F. plautii* ATCC 49531 (Fig. 1b), based on the following hypothesis: functionally associated genes often exhibit the co-location (remaining in close physical proximity to each other), co-occurrence (both occurring or not occurring), or fusion pattern in the genome[21,22]. The yielded 14 genes were then expressed in *E. coli* BL21, and the resulting transformants were cultured in the medium supplemented with apigenin. Encouragingly, apigenin consumption and naringenin production were observed in the culture supernatant of the *E. coli* strain expressing the A4U99_05915 gene (coding for the KGF53654.1 protein) (Fig. 1c and Supplementary Fig. 2), whereas no obvious change occurred for the other strains (Supplementary Fig. 2). These results indicate that the A4U99_05915 gene encodes an enzyme capable of converting apigenin to naringenin, which is likely to be our target enzyme responsible for the hydrogenation of the C2–C3 double bond of apigenin in *F. plautii*. Here, we named this enzyme flavone reductase (FLR).

**Characterization of FLR reveals its specific activity towards flavones and flavonols.** To elucidate the catalytic property of FLR, we expressed and purified this enzyme for in vitro assays. Since the purified FLR had characteristic absorption peaks at 373 and 449 nm in a full-wavelength scan (Supplementary Fig. 3), which are the same as those of flavin, we speculated that FLR was co-purified with flavin mononucleotide (FMN) or flavin adenine dinucleotide (FAD), two cofactors widely distributed in nature. Therefore, a reaction system coupling NADH consumption and $FMNH_2$ generation was designed to assay FLR activity towards apigenin (Fig. 2a). As expected, decrease in apigenin level and

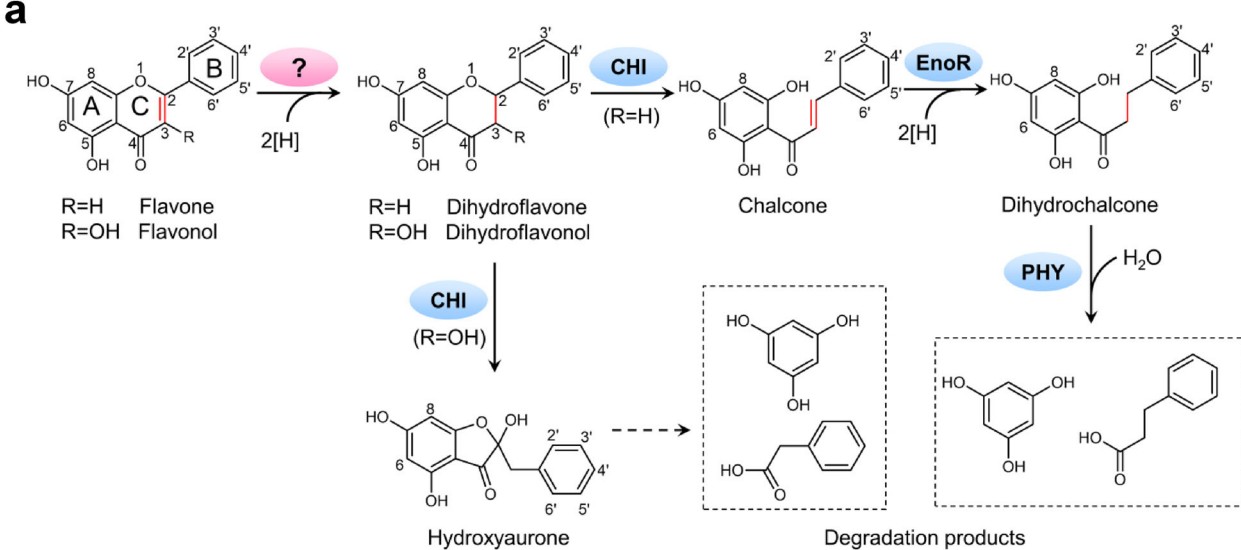

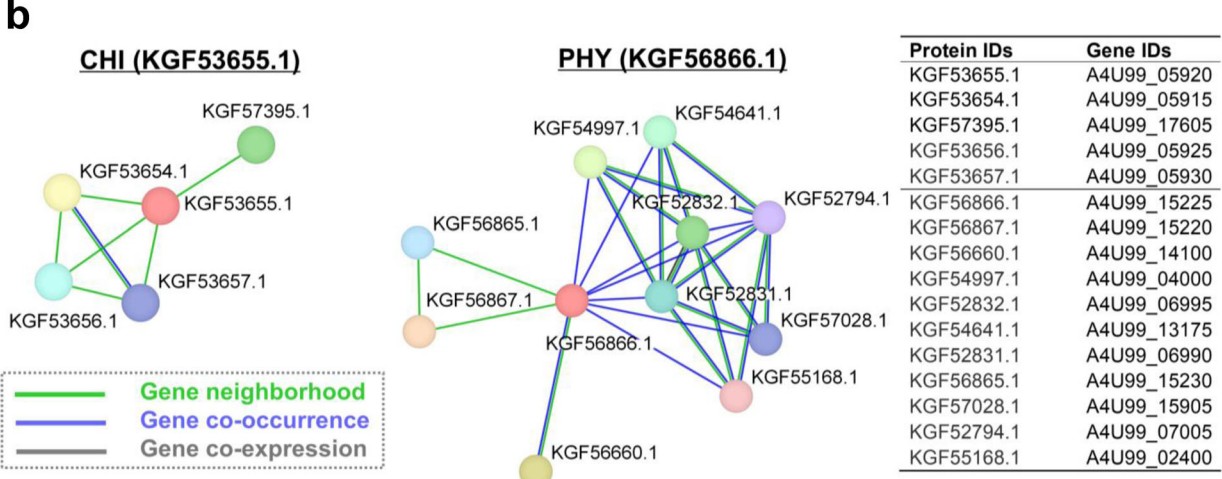

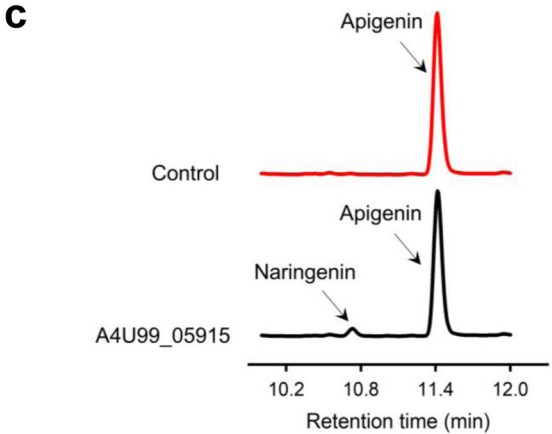

**Fig. 1 Discovery of the flavone reductase (FLR) in *F. plautii* ATCC 49531. a** Proposed microbial metabolic pathway for flavones and flavonols. Enzymes that were identified are indicated with blue ellipses. The question mark in the pink ellipse indicates that the enzyme responsible for this step is missed. CHI, chalcone isomerase; EnoR, enoate reductase; PHY, phloretin hydrolase. **b** Fourteen potential FLR enzymes generated by the comparative genomic analysis of *F. plautii* ATCC 49531. **c** HPLC detection of the conversion of apigenin to naringenin by the *E. coli* strain (BL21) expressing the A4U99_05915 gene (coding for the KGF53654.1 protein from *F. plautii* ATCC 49531).

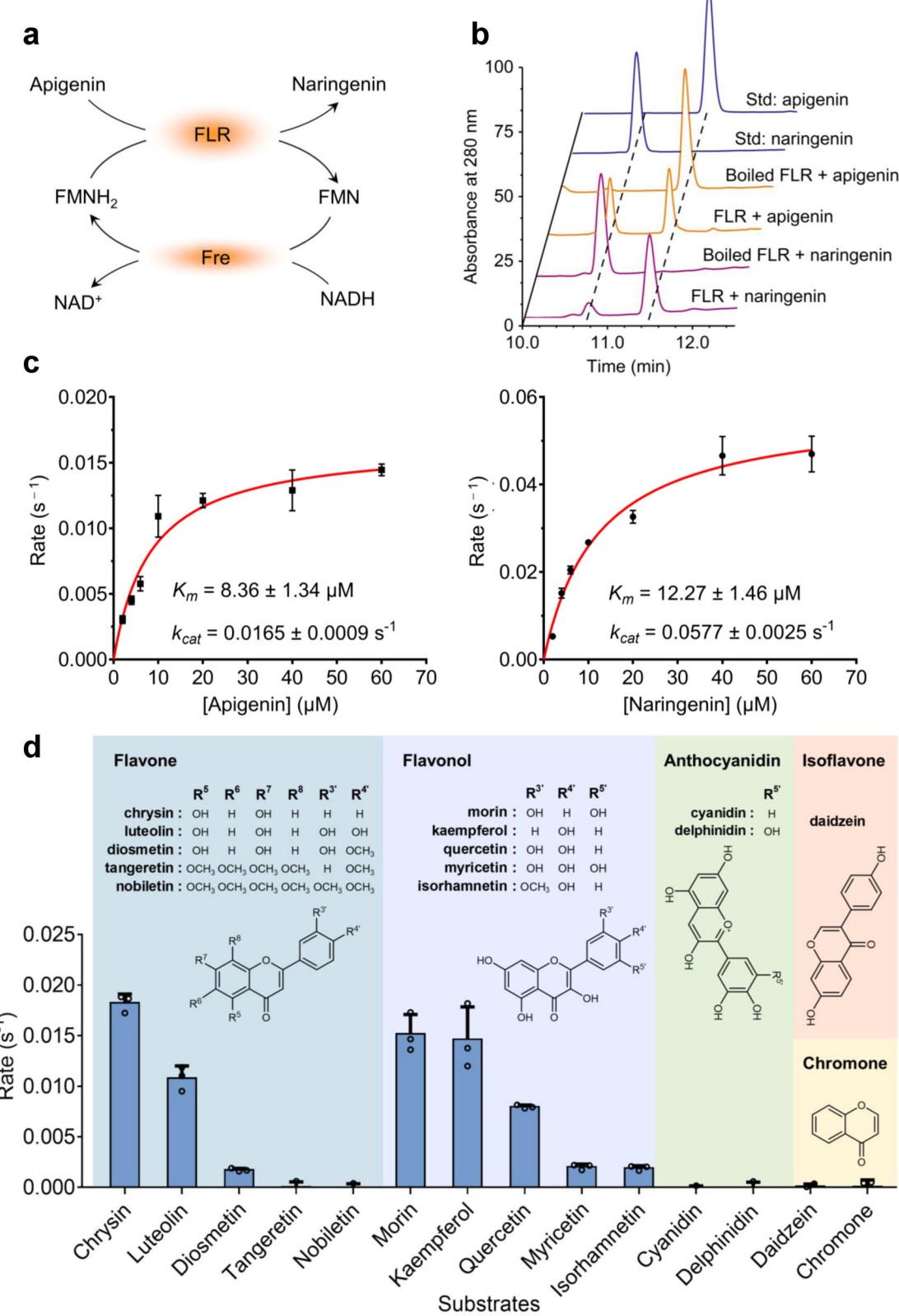

**Fig. 2 Functional identification of flavone reductase (FLR). a** The reaction system, coupling NADH consumption and FMNH$_2$ generation, for the assay of FLR activity towards apigenin. Fre, NADH-specific FMN oxidoreductase. **b** HPLC analysis of the products formed by the in vitro FLR-catalyzed conversion between apigenin (Api) and naringenin (Nar). The control reactions were carried out using the disabled FLR (boiled). **c** In vitro steady-state kinetic analysis of FLR using apigenin or naringenin as the substrate. Each data point represents mean ± SD ($n = 3$). Error bars show SDs. **d** A screening of the substrate spectrum of FLR. Initial turnover rates of purified FLR towards different substrates (100 μM) were measured. Data are represented as mean ± SD ($n = 3$). Error bars show SDs.

production of naringenin were observed in the presence of FLR (Fig. 2b). The liquid chromatography-mass spectrum (LC-MS) analysis further confirmed the formation of naringenin (with the molecular ion fragment of $m/z$ 273.1) (Supplementary Fig. 4a–c). Interestingly, when NADH was replaced by $NAD^+$ in this coupled reaction system (Fig. 2a), FLR could reversely convert naringenin to apigenin (Supplementary Fig. 4a, d, e). The kinetic parameters of FLR in transforming apigenin and naringenin were also assayed under optimal reaction conditions (37 °C and pH of 7.0) (Supplementary Fig. 5), showing a higher $k_{cat}/K_m$ value towards the latter (Fig. 2c).

To further define FLR's substrate scope, we assayed FLR's activity towards other common dietary and medicinal flavones, including chrysin, luteolin, diosmetin, tangeretin, and nobiletin. The results showed that FLR has high activity towards chrysin, luteolin, and diosmetin, but no activity towards tangeretin and nobiletin, which both contain multiple methoxyl groups on the A and B ring (Fig. 2d).

Given the highly similar structure of flavonols and flavones (Fig. 1a), we further assessed FLR's activity towards five representative flavonols, i.e., morin, kaempferol, quercetin, myricetin, and isorhamnetin. Interestingly, FLR displayed activity towards all these compounds (Fig. 2d), although the reaction rate gradually decreased (morin > kaempferol > quercetin > myricetin > isorhamnetin) with the increased number of hydroxyl or methoxyl groups on the scaffold (Fig. 2d). However, FLR did not display activity towards other structurally similar flavonoids, including isoflavone (daidzein) and anthocyanidin (cyanidin and delphinidin) (Fig. 2d). In addition to the in vitro biochemical analysis, the in vivo function of FLR in flavonoid metabolism in gut bacterium was also confirmed by the genetic deletion experiments, and the detailed data will be shown later. Altogether, these results strongly suggest that FLR's activity is restricted to flavones and flavonols, and simultaneously, can tolerate differences in the number and position of hydroxyl and methoxyl groups on the substrate skeleton to some extent. To our knowledge, such a catalytic property towards flavones and flavonols has not been previously discovered for other known reductases.

**The widespread FLR-like enzymes are a distinct class of ene-reductases**. We next investigated the distribution of FLR-like enzymes in microbial communities using the FLR sequence of *F. plautii* ATCC 49531 as the query. The BLASTP search identified 391 putative FLR enzymes (amino acid sequence identity ≥30%, coverage ≥80%, e-value ≤1e−10) from over 150 microbial species (including some unclassified species), suggesting that these enzymes are widespread in microbes. To further elucidate their relationship, we carried out a maximum-likelihood phylogenetic analysis of 72 putative FLR enzymes from 54 microbial strains (Supplementary Table 1) that have complete genome data in KEGG database. The results revealed that most of the FLRs exist in prokaryotes, in which *Firmicutes* bacteria, a predominant group of gut bacteria, occupied the largest proportion (Fig. 3a). FLRs were also found in a few *Fusobacteria*, *Actinobacteria*, and *Spirochetes* bacteria, some of which are pathogens or commensal bacteria normally colonized in the digestive tract (Fig. 3a). To further verify the phylogenetic analysis, 11 putative FLRs in different isolates, together with the aforementioned *F. plautii* FLR, were picked out to examine their activity towards the flavone apigenin. As expected, the conversion of apigenin was observed for all these enzymes (Supplementary Fig. 6), indicating that the FLR-like enzymes present in the phylogenetic analysis are reliable.

The enzymes characterized to date that are capable of catalysing asymmetric activated alkene reduction are assigned as ene-reductases[23], including enoate reductase (EnoR) and isoflavone reductase (IFR) (Fig. 1a), which are involved in flavonoid transformation. The FLR enzyme identified in this study, based on its catalytic property, is also an ene-reductase; however, our BLASTP analyses indicated that FLR has a remote revolutionary relationship with currently known ene-reductases (Supplementary Table 2). To systematically assess the relationship between the FLR-like enzymes and identified ene-reductases, we constructed two sequence similarity networks (SSNs) using the aforementioned 72 FLR sequences and almost 2000 sequences belonging to the known five ene-reductase families from the UniProtKB protein database (Fig. 3b and Supplementary Fig. 7). As expected, the results of SSN analysis suggest that FLRs are a highly distinct group of ene-reductases that have not yet been characterized. Additionally, FLR was found incapable of directly using NAD(P)H as the reducing agent (Supplementary Fig. 8), and thus, is not an NAD(P)H-dependent reductase, which is different from known ene-reductase[23]. Therefore, a complete electron transfer chain for FLR remains to be explored.

**Crystal structure analysis of FLR**. To elucidate the detailed catalytic mechanism of FLR, we solved the crystal structure of FLR by using the Se-SAD method (Supplementary Table 3). The overall structure of FLR consists of an N-terminal compact α-β-α domain (residues 1–202) and a C-terminal extended domain (residues 222–308), connected by a long loop (Fig. 4a). The C-terminal domain can be further divided into subdomains 1 and 2; the former is comprised of 4 β-strands, while the latter contains a bended long helix. Two symmetry-related FLR monomers form a homodimer (Fig. 4b). The dimer interface includes both the N-terminal and C-terminal domains, burying 29.5% (4540.3/15381.8 $Å^2$) surface area. Specially, the C-terminal subdomain-1 from both monomers pack against each other forming a two-layered β-sheet, while the subdomain-2 or bended long helix from each monomer adopts an encircled-arm conformation holding the N-terminal domain of the neighbor molecule. The cofactor molecule FMN binds at the dimer interface (Fig. 4b). Additionally, we observed that the FLR proteins of solution state mainly exist as dimers, as determined by size-exclusion chromatography coupled with multi-laser light scattering (Supplementary Fig. 9). All these findings strongly suggest that the homodimer is the functional unit of FLR.

FLR shows catalytic activity towards various flavonoids. To further illuminate the underlying mechanism, we solved the complex structures of FLR with various substrates, including apigenin, chrysin, and luteolin that contain similar A/C rings but differ in the B ring. The structures were refined to 2.55, 2.65, and 2.25 Å resolution, respectively, which adopt a very similar conformation with r.m.s.d. value of 0.18~0.21 Å (Supplementary Table 3). The substrate and the cofactor FMN bind in the same pocket, which is located at the dimer structure interface and mainly comprised of the N-terminal domain from one molecule and the C-terminal subdomain-2 from the neighbor molecule. The structures clearly demonstrate that the coordination of substrate mainly involves the A/C rings, while the B ring protrudes out of the pocket. Specifically, the A/C rings of substrates stack against the isoalloxazine ring of FMN on one side and the side chain of residue Ile275 of subdomain-2 on the other side. This substrate-binding pattern may explain why FLR can catalyze the reaction of apigenin, chrysin, and luteolin, since all these three substrates differ only in the B ring. (Supplementary Fig. 10, lower panels).

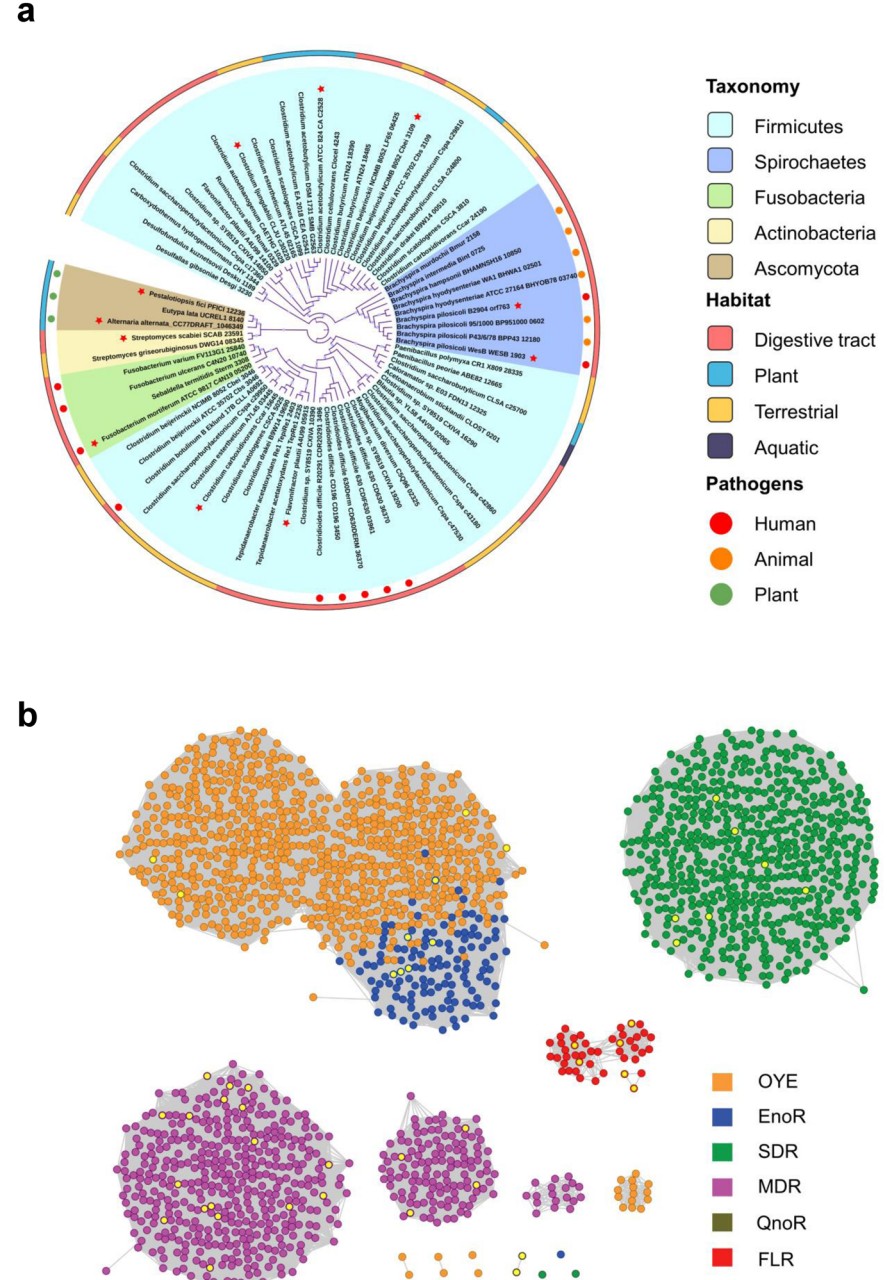

**Fig. 3 Bioinformatic and experimental analysis identify that flavone reductases (FLRs) are a distinct group of ene-reductases. a** Phylogenetic analysis of FLRs. The maximum-likelihood phylogenetic tree is based on the protein sequences of putative FLR enzymes found in Ascomycota, Firmicutes, Spirochetes, Fusobacteria, and Actinobacteria. Light purple circles on branches indicate bootstrap values greater than 0.7 from 300 bootstrap replicates. The FLR-like enzymes highlighted with red stars were chosen to assay the in vitro activity towards the flavone apigenin. **b** A sequence similarity network (SSN) for the ene-reductases. The network was generated with an initial score of $10^{-20}$ and the "alignment score threshold" (that is a measure of the minimum sequence similarity threshold) for drawing the edges that connect the proteins (nodes) in the SSN was then refined such that nodes are connected by an edge if this value is ≥40. Each of the nodes represents proteins with ≥90% amino acid sequence identity and was colored according to the cluster type (OYE, EnoR, SDR, MDR, QnoR, and FLR). Each type of ene-reductase, as shown in a different color, is separated into different clusters that may contain enzymes with similar biochemical activity. A total of 72 different FLR-like proteins which are shown in the phylogenic tree are denoted in red. The nodes representing the reported ene-reductase with enzyme activity data (from BRENDA Enzyme Database) and biochemically identified FLR-like enzymes in this study were highlighted with lemon yellow. OYE, Old-Yellow-Enzyme; EnoR, oxygen-sensitive enoate reductases; SDR, short-chain dehydrogenase/reductase; MDR, medium-chain dehydrogenase/reductase; QnoR, quinone reductase-like ene-reductase.

Next, we used the structural information derived from apigenin-FMN-FLR complex for verification (Fig. 4c). As demonstrated by the structure, the phosphate of FMN forms four coordinates with residue Met10 from the loop connecting β1 strand and α1 helix, and residues Ser14 and Asn15 from the α1 helix. The ribitol of FMN forms three hydrogen bonds with residue Val144 from the β4 strand, and residue Ile89 from the loop connecting β3 strand and α4 helix. The isoalloxazine of FMN forms at least ten coordinates with residues Gly146, Ser147, and Trp149 from the loop connecting the β4 strand and α7 helix,

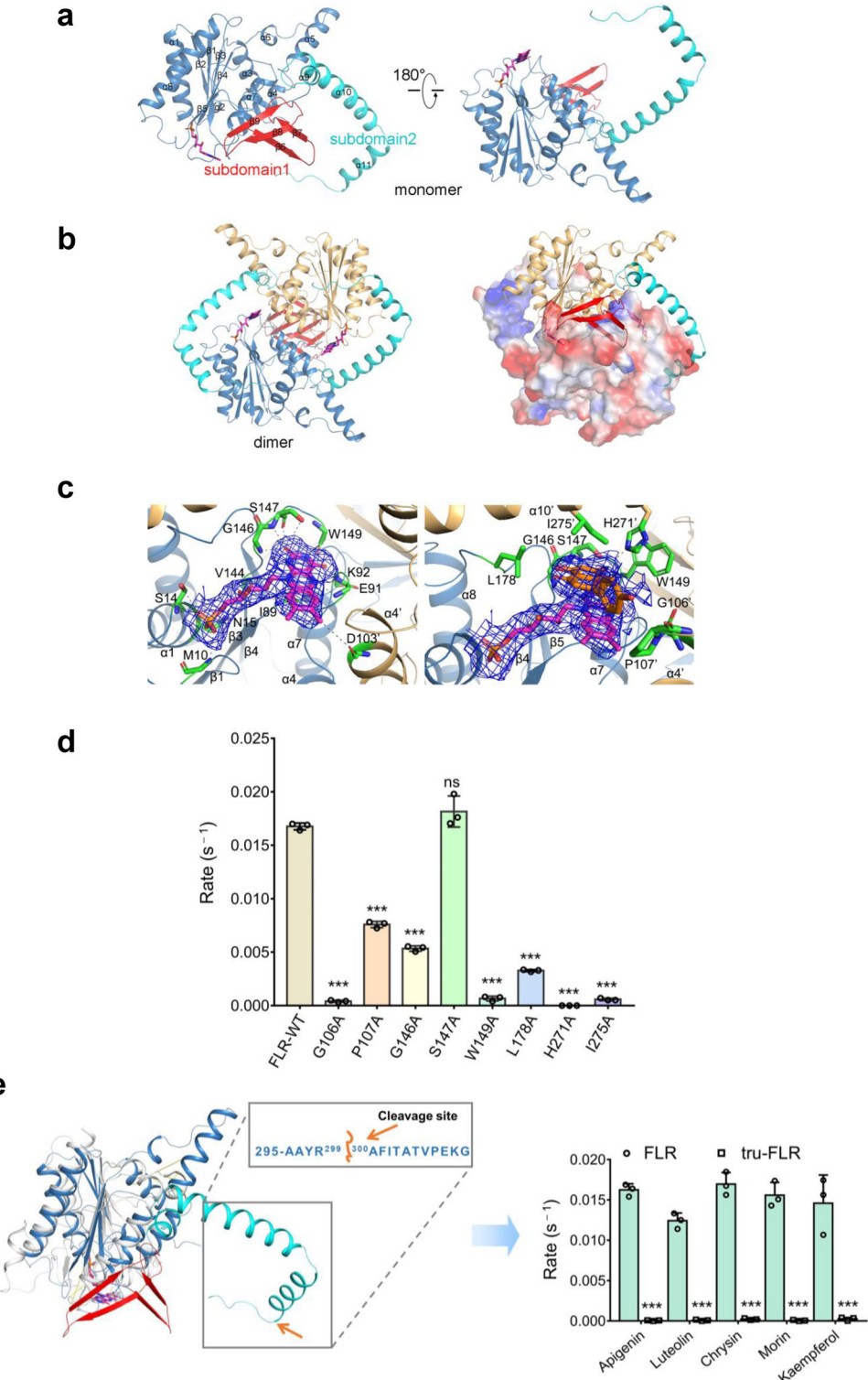

**Fig. 4 Crystal structure of flavone reductase (FLR). a** Crystal structure of FLR monomer. The N-terminal domain, and C-terminal subdomains 1 and 2 are colored slate-blue, red, and cyan, respectively. FMN is shown with magenta sticks. **b** Crystal structure of FLR dimmer and its interface. Monomers A and B are shown with ribbon cartoon and electron static surface potential, respectively. **c** Zoom-in views of the cofactor FMN- and apigenin-binding site. The pictures show the amino acid residues surrounding the active site, and Fo-Fc density map (blue), contoured at 1.0 σ level, from which the FMN and apigenin were omitted. **d** In vitro enzymatic activity assay (initial turnover rate) of the wild-type (WT) or mutants of FLR. Data are represented as mean ± SD ($n =$ 3). Error bars show SDs. Statistical analysis was performed by a two-tailed Student's $t$-test. [***]$P < 0.001$ versus FLR-WT. ns, not significant. **e** Structural superimposition of the FLR monomer with the WrbA protein (PDB:3B6I, colored gray). Tru-FLR, the truncated FLR protein. Initial turnover rates of purified FLR and Tru-FLR towards different substrates were measured. Data are presented as mean ± SD ($n = 3$). Error bars show SDs. Statistical analysis was performed by a two-tailed Student's $t$-test. [***]$P < 0.001$ versus FLR.

residues Ile89, Glu91, and Lys92 from the loop connecting the β3 strand and α4 helix, and residue Asp103 from the α4 helix of another monomer. Apigenin forms two coordinates with residue Ser147 from the loop connecting the β4 strand and α7 helix and residue His271 from the α10 helix of another monomer (Fig. 4c). The significance of residues involving in the binding of apigenin was verified by enzymatic assay. The results show that the reduction activities of most mutants were reduced, among which those of G106A, W149A, H271A, and I275A mutants were significantly reduced, or even eliminated (Fig. 4d), and surprisingly, the reduction activity of S147A mutant is slightly higher than that of wild-type (WT) FLR. These data confirm the roles of residues involving substrate and cofactor coordination. Together, our structural data well explain the substrate selectivity and catalytic promiscuity of FLR.

Interestingly, a detailed search using the SWISS-MODEL tool revealed that FLR has a structural fold similar with that of the WrbA protein (r.m.s.d. = 1.8 Å) from *E. coli* K12, except that FLR has an extra protruding "tail" at the C-terminal (approximately 310 amino acid residues) (Fig. 4e). WrbAs are a large group of oxidoreductases that function in response to environmental stress[24,25]. Based on the structural model, we reasoned that such an extra "tail" of FLR compared to WrbA may be essential for FLR's specific catalytic activity. From the FLR-apigenin complex structure (Supplementary Fig. 11), we observed: (i) the A/C rings of apigenin stack against the isoalloxazine ring of FMN by π–π interactions on one side and the hydrophobic side chain of the residue Ile275 on the other side; (ii) the His271 forms a hydrogen bond with the hydroxyl group at the C-7 position of the A ring of apigenin; (iii) the residue Lys282 of tail interacts with FMN through a water molecule; (iv) the residue Thr305 of tail forms a hydrogen bond with Glu34 of another monomer at the interface; (v) the aromatic ring system is found to stack between the tetrahydropyrrole ring of Pro307 and the indole side chain of Trp72 of another monomer. Together, these findings indicate that the tail is involved not only in substrate binding, but also in dimer stabilization. To confirm this speculation, a truncation of this C-terminal domain was performed. As expected, the deletion of 11 amino acid residues completely abolished the enzyme activity of FLR towards flavones and flavonols, including apigenin, luteolin, chrysin, morin, and kaempferol (Fig. 4e). Meanwhile, we proved that the WrbA protein from *E. coli* (K12) has no catalytic activity on the flavone apigenin (Supplementary Fig. 12). Therefore, these data demonstrate the importance of this C-terminal "tail" for FLR.

**FLR is crucial for the metabolism and utilization of dietary and medicinal flavones/flavonols.** Having determined the in vitro activity of FLR, we next sought to examine whether this ene-reductase functions in gut microbes. Because *F. plautii* ATCC 49531 cannot be genetically manipulated yet, another gut bacterium, *Clostridium ljungdahlii* DSM 13528[26], was used for the test. We first searched for putative *flr* genes in the *C. ljungdahlii* genome using the BLASTP tool. As expected, a potential FLR protein was found, which is encoded by the Clju_c 30220 gene (Fig. 5a) and shows a significant homology (e-value 5e⁻³³) as well as moderate amino acid identity (29.69%) to the above-mentioned FLR protein of *F. plautii* ATCC 49531. Two other proteins involved in flavone metabolism (EnoR and PHY) were also found, showing 34.49% and 53.23% amino acid sequence identity to the homologs in *F. plautii* ATCC 49531, respectively (Fig. 5a). These data suggest that *C. ljungdahlii* DSM 13528 has potential for transforming flavones and flavonols. Next, we deleted the Clju_c 30220 gene in the WT *C. ljungdahlii* DSM 13528 (WT *C. ljungdahlii*), which resulted in a mutant strain (*C.*

*ljungdahlii* Δ*flr*). The growth curves of these two strains showed no obvious differences (Supplementary Fig. 13), indicating that the deletion of the Cjlu_c 30220 gene does not affect the normal growth of *C. ljungdahlii*.

Then, the A4U99_05915 gene encoding the FLR protein of *F. plautii* ATCC 49531 was introduced into the *C. ljungdahlii* Δ*flr* strain via the plasmid pMTL83151, which yielded a complemented mutant *C. ljungdahlii* Δ*flr* p*flr*. Next, the WT *C. ljungdahlii*, *C. ljungdahlii* Δ*flr*, and *C. ljungdahlii* Δ*flr* p*flr* strains were cultivated in a defined medium supplemented with various flavone or flavonol compounds (apigenin, quercetin, diosmetin, and baicalein) to compare their utilization of these substrates, aiming to examine the in vivo role of FLR in microbial metabolism of dietary or medicinal flavones and flavonols. Here, apigenin and quercetin are the representative dietary flavone and flavonol, respectively, while both diosmetin and baicalein are medicinal flavones. The results showed that the WT *C. ljungdahlii* strain could consume all the apigenin, quercetin, and diosmetin as well as about half of the baicalein within 48 h (Fig. 5b). In contrast, the *C. ljungdahlii* Δ*flr* strain did not consume these substrates; however, this phenotypic defect could be restored in the *C. ljungdahlii* Δ*flr* p*flr* strain (Fig. 5b). These results demonstrate that FLR is indeed essential for flavone and flavonol metabolism in *C. ljungdahlii* DSM 13528, which may be also extended to other flavone/flavonol-metabolizing gut microbes.

**FLR affects the microbial community composition and is widespread in the gut microbiome.** A question raised from the above findings is whether the FLR-initiated metabolism of flavones and flavonols can endow the microbial hosts with growth advantage over those *flr*-missing gut microbes, thereby affecting the gut microbial community composition. To answer this question, we constructed a simplified gut microbiota consisting of eight representative gut bacterial strains (containing no putative FLR homologs). This simplified gut microbiota was then co-cultivated with the aforementioned WT *C. ljungdahlii* or *C. ljungdahlii* Δ*flr* strain in a defined medium (with 1 mM of apigenin) to compare their bacterial community composition at different time points. As shown in Fig. 6a, the WT *C. ljungdahlii* strain exhibited an obvious growth advantage over the other co-cultivated bacteria at the 8 h time point, but such a phenomenon was not detected for the *C. ljungdahlii* Δ*flr* strain. In contrast, the other strains had no significant changes in their percentages in the microbial community. Furthermore, the effect of apigenin on the growth of each individual isolate in this microbial community was also examined. The results showed that the growth of all isolates, except *C. ljungdahlii* (WT) and *Bifidobacterium bifidum*, were subjected to different degrees of inhibition under apigenin stress (Supplementary Fig. 14). On the contrary, the growth rate of the WT *C. ljungdahlii* strain in the presence of apigenin was faster than that without apigenin (Supplementary Fig. 14). Overall, these results indicate that FLRs may make contributions to the growth of the microbial hosts in the presence of flavones/flavonols, which is probably due to their utilization of flavones and flavonols as the carbon and energy sources as well as the derived tolerance to these antibacterial compounds[27].

We further investigated whether FLR-like enzymes are prevalent and abundant in the human gut microbiota, using the MetaQuery (http://metaquery.docpollard.org/) database[28,29], which contains data on >2000 human gut metagenomes. The result showed that the putative *flr* genes are abundant in the human gut microbiota, mainly ranging from the abundance of 0.001 (1 copy per 1000 cells) to 0.1 (1 copy per 10 cells) (Fig. 6b). The average estimated copy number of this gene is 1.2 per 100 cells (Fig. 6b). We also assessed the prevalence of the *flr* genes

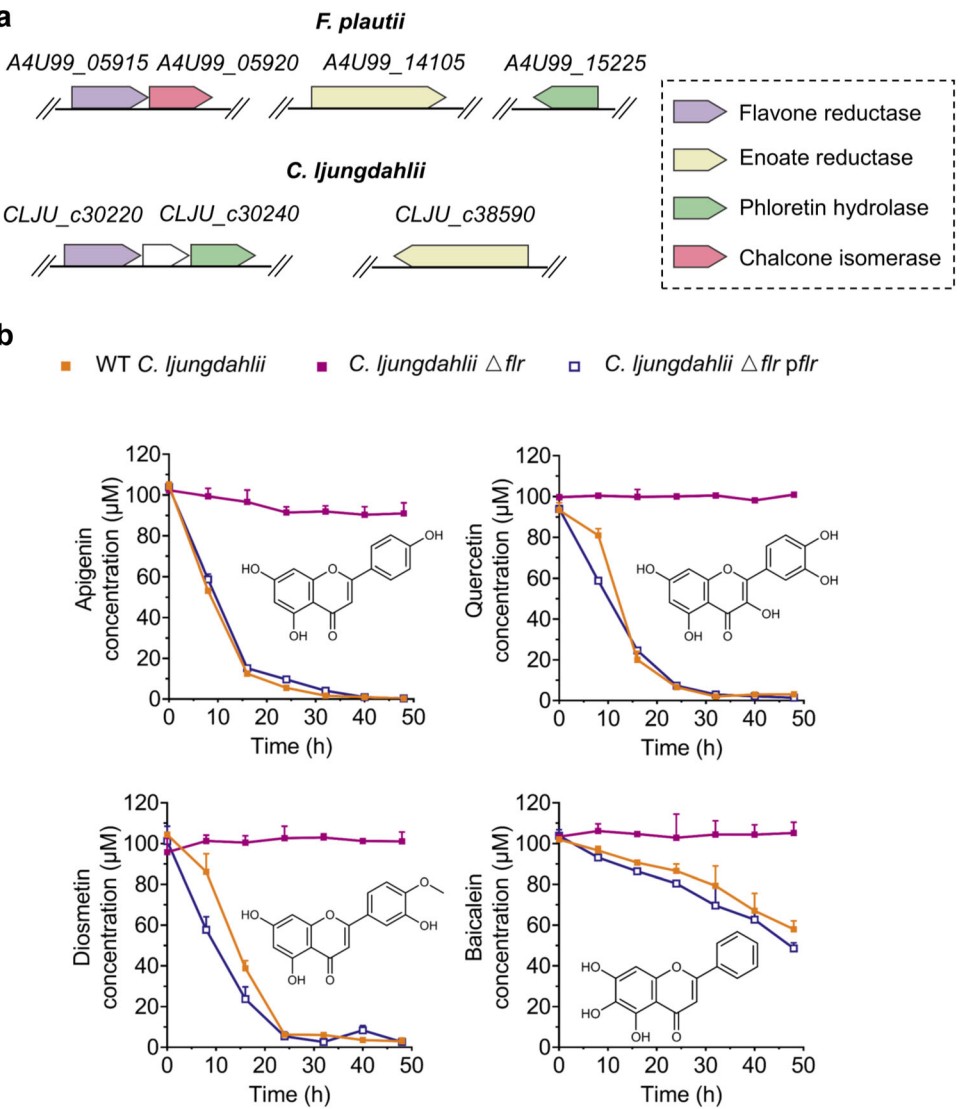

**Fig. 5 Significance of flavone reductase (FLR) in the clostridial metabolism of dietary or medicinal flavones and flavonols. a** Homologous genes involved in the degradation of flavones and flavonols in *F. plautii* ATCC 49531 and *C. ljungdahlii* DSM 13528. Homologous genes are shown with same colors. **b** The consumption of apigenin, quercetin, diosmetin, and baicalein by the WT *C. ljungdahlii* (WT), *C. ljungdahlii Δflr* (*flr* deletion), and *C. ljungdahlii Δflr pflr* (genetic complementation) strains. The WT *C. ljungdahlii* strain and its derived strains were grown in the YTF medium, which contained 10 g L$^{-1}$ of fructose as the extra carbon source. Data are presented as mean ± SD (*n* = 3). Error bars show SDs.

across human gut metagenomes at different abundance thresholds and found that these genes occurred in 90.5% samples at a minimum abundance of 0.001(1 copy per 1000 cells) (Fig. 6c). These data suggest that FLR-like enzymes are widespread in the human microbial population. Whether these enzymes are necessarily involved in the gut microbial flavonoid degradation needs further experimental verification.

## Discussion

Gut microbial metabolism of flavonoids has been extensively linked to the bioavailability of these natural plant products to humans. However, we currently still have a relatively poor understanding on the specific genes and enzymes that mediate gut microbial activities. This study reveals, to our best knowledge, a previously unknown class of ene-reductases (FLRs) that demonstrate exceptional catalytic selectivity towards flavones and flavonols, thereby filling a key gap in the microbial metabolic pathways of flavonoids. The functional and physiological

importance of FLR in the human gut microbiota was also revealed, expanding our knowledge of the contribution of ene-reductase families to humans.

Thus far, the representative ene-reductases involved in flavonoid metabolism include enoate reductase (EnoR) and isoflavone reductase (IFR)[19,30], belonging to the ene-reductase class of EnoR and short-chain dehydrogenase/reductase (SDR), respectively, and both are capable of using NAD(P)H as the reducing agent[23]. In contrast, the FLR enzyme identified in this study is not an NAD(P)H-dependent ene-reductase. Despite the substrates (iso-flavones) of IFR having structures highly similar to those (fla-vones) of FLRs (Supplementary Fig. 1a), we did not observe any intersection in the substrate scopes of these two enzymes. Based on the co-crystal structures of FLR and its substrates, such a distinct substrate spectrum of FLR is likely due to the combined effect of the specific residues lining its substrate binding pocket. In addition, although the *E. coli* WrbA protein has a structural superimposition with the FLR monomer (Fig. 4e), the BLASTP search revealed that these two proteins have low similarity (31%

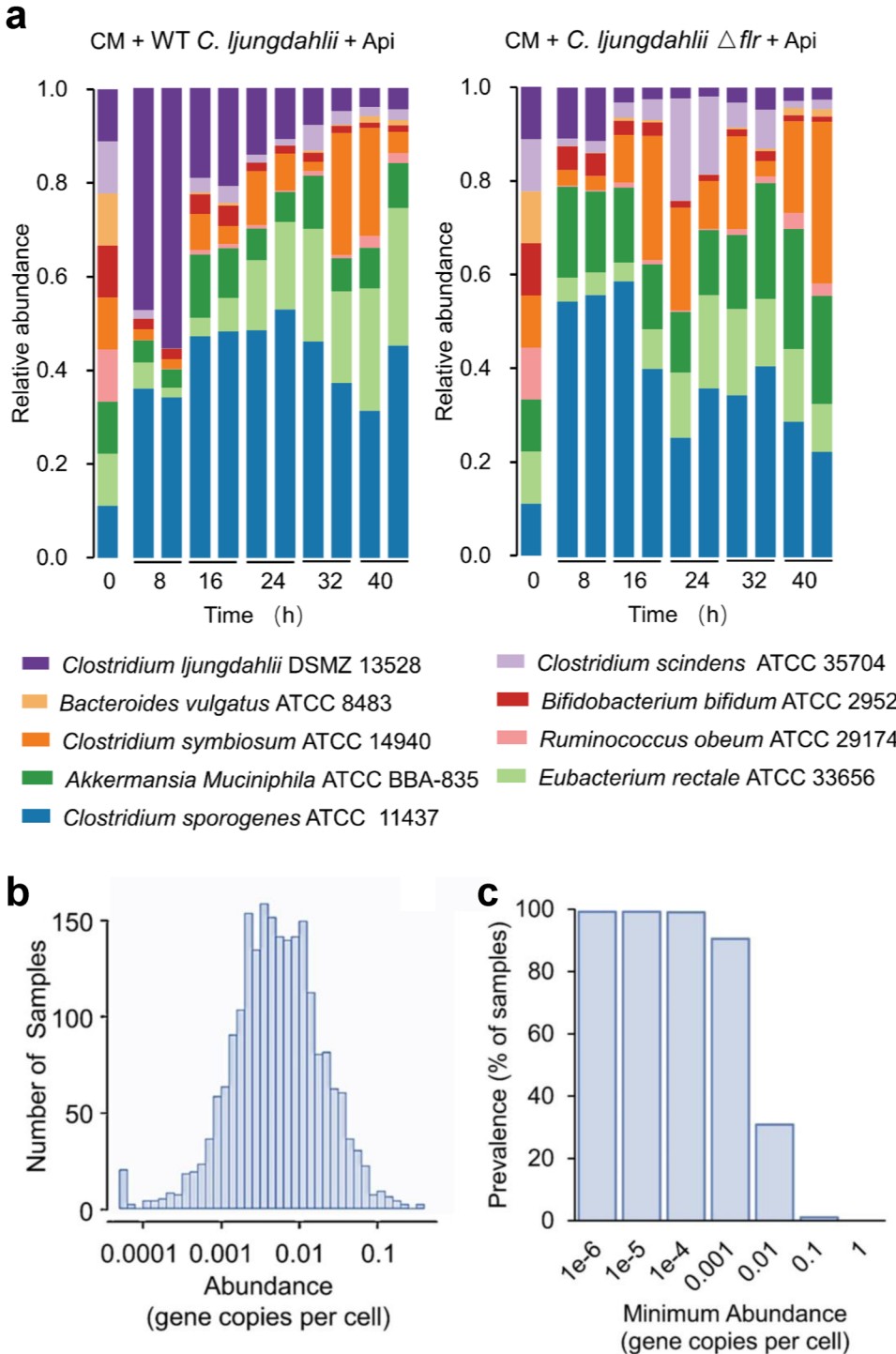

**Fig. 6 Influence of the *flr* gene on gut microbial composition under the pressure of flavone apigenin and the abundance of *flr*-like genes in human gut microbiome. a** Comparison of relative abundance of the WT *C. ljungdahlii* (WT) and *C. ljungdahlii* Δ*flr* (*flr* deletion) strains in a simplified gut microbiota (containing eight representative human gut bacterial strains without putative *flr* genes) under the pressure of apigenin. At the beginning of this experiment, the same volume (2 mL) of the cultures of these strains (OD$_{600}$ = 0.8) were mixed; then, 20 mL of the mixture was added aseptically to 200 mL of GMM broth containing 1% mucin and 1 mM apigenin. Therefore, the relative abundance of each strain in this artificial microbial community was considered to be the same at 0 h. The abundance of *groEL* (marker gene) at different time points relative to that at 0 h was used to indicate the relative abundance of individual strain. CM, simplified core microbiota consisting of eight representative gut bacterial strains. Api, apigenin. The data of two biologically independent samples were presented at each time point. **b**, **c** The abundance of the *flr* homologs across human gut microbiome samples (**b**) and the prevalence of the *flr* homologs across human gut metagenomes at different abundance thresholds (**c**) based on the human microbiota metagenomic data from MetaQuery.

amino acid sequence identity, 31% coverage, and e-value 9e−03). It is known that flavone synthase I (FNSI) and II (FNSII), the two plant enzymes that belong to the 2-oxoglutarate-dependent dioxygenases and cytochrome P450 monooxygenases, respectively, can catalyze flavone formation from dihydroflavone[31]; however, such a $O_2$-dependent reaction is generally irreversible[32,33]. Thus, FNSI and FNSII are incompetent for the FLR's role in reducing flavones. Together, these data and analysis suggest that the FLR enzyme is distinct from all the existing reductases or flavone synthases.

As an enzyme of reversible reaction, FLR can also catalyze the flavone formation from dihydroflavone efficiently. As shown in Fig. 2c, the $k_{cat}/K_m$ value of FLR towards naringenin is much higher than that towards apigenin, and even comparable to those for some existing flavone synthases[34,35]. This suggests that FLR and its homologs have the potential in the construction of artificial pathways for flavone production in heterologous microbial hosts, e.g., E. coli. Although there are several enzymes from plants known to be used in microbial hosts for transforming dihydroflavones to flavones[36], these, however, normally have poor compatibility with the microbial hosts without artificial modifications[37]. Such a problem is expected to be effectively resolved by using microbial enzymes, but until now, no responsible enzyme of this reaction step has been identified in microorganisms. Therefore, the FLR-like enzymes discovered from microorganisms provide a valuable reservoir of genetic resources for biocatalytic applications.

Interestingly, structural analysis revealed that the binding pocket of FLR has two major differences from those of Old-Yellow-Enzymes (OYEs): (1) FLR is devoid of a suitable residue (such as the His181 and the His184 residues in the pentaerythritol tetranitrate reductase of OYE families) that can form a direct hydrogen bond to the C4-carbonyl group of apigenin for substrate activation[38]. Instead, our complex structure revealed that a hydrogen-bond relay was formed from the hydroxyl of Thr279 through $H_2O$ to the C4-carbonyl group of apigenin (Supplementary Fig. 15a), probably leading to substrate activation. (2) FLR also lacks a direct proton donor to the α-carbon of apigenin for reduction, which is different from OYEs that use tyrosine residue as a general acid[38]. However, we observed that the α-carbon is exposed to the bulk solvent, and thus the proton required is likely derived from water molecules. Based on above findings, a possible catalytic mechanism for FLR was proposed (Supplementary Fig. 15b). Briefly, after substrate activation induced by Thr279, the hydride from FMNH⁻ is transferred to the β-carbon of apigenin and the α-carbon obtains a proton from water solvent, thereby forming the C2-S-naringenin product, which was supported by the analysis of circular dichroism (CD) spectrum (Supplementary Fig. 16).

Our study suggests that gut clostridia, and probably many more other microorganisms, metabolize flavones and flavonols using FLR enzymes. We found that most of these microbes live either in the intestinal tract or around plants, where flavonoid compounds are normally abundant, indicating that FLR enzymes may have evolved for degrading these plant products. From an ecological perspective, FLR-mediated bioconversions can protect some microbial hosts from the toxicity of flavone and flavonol compounds, thereby giving these microbial individuals a growth advantage over the others in their microbial community. Our results also support this speculation (Fig. 6a). However, just as a coin has two sides, such a metabolic capacity will also protect some pathogens from the antibacterial effects of flavones and flavonoids. For example, BLASTP search has revealed definitive copies of the FLR-encoding gene (flr) in some pathogenic gut bacteria, including Clostridium difficile DSM 630, Clostridium botulinum BKT015925, Clostridium perfringens ATCC 13124,

and Brachyspira pilosicoli B2904, suggesting that these pathogens may have the ability to transform flavones and flavonols for self-protection.

The analysis of human microbiota metagenomic data from MetaQuery confirms that the flr-like genes are not only broadly distributed, but also richly present in the human gut microbiome (Fig. 6b, c). Human intestinal tract hosts thousands of microbial species, and the prevalence of the flr genes in this microbial community may be due to the horizontal gene transfer. However, the factors that determine the specific distribution of these genes in the microbial community remain unclear. Nevertheless, considering that a large number of flavone and flavonol compounds have been used as clinical drugs[39,40], it seems imperative to probe the potential impact of gut microbial transformations on the efficacy of these drugs. The flr genes may be useful predictive biomarkers for assessing the flavone and flavonol drug transformation in different individuals, thereby guiding personalized medicine.

## Methods

**Bacteria, media, reagents, and growth conditions.** The E. coli DH5α and its derived strains were grown in LB (lysogeny broth) medium[41] supplemented with chloramphenicol (12.5 μg mL⁻¹) when needed. The E. coli BL21 and its derived strains were also grown in LB medium supplemented kanamycin (100 μg mL⁻¹) when needed. The F. plautii ATCC 49531 strain was grown in Wilkens-Chalgren Anaerobe (WCA) Broth[42]. The C. ljungdahlii strain (DSM 13528) was grown in YTF medium[43], in which 5 μg mL⁻¹ of thiamphenicol was added as needed for plasmid selection. Anaerobic cultivation of bacteria was performed in an anaerobic chamber (Whitley A35 Anaerobic Workstation, Don Whitley Scientific Limited, Bingley, West Yorkshire, UK).

Commercial chemicals were purchased from Sigma-Aldrich (Sigma-Aldrich Co., St Louis, USA) and Aladdin (Shanghai Aladdin Biochemical Technology Co., Ltd., Shanghai, China). KOD plus Neo and KOD FX DNA polymerase (Toyobo, Osaka, Japan) were used for PCR amplification. The primers used in this study were synthesized by GenScript (GenScript, Nanjing, China). The restriction enzymes and ligase used in plasmid constructions were purchased from Thermo Fisher Scientific (Thermo Fisher Scientific, Vilnius, Lithuania) and Takara (Takara, Dalian, China), respectively. The assembly of multiple DNA fragments in plasmid construction was performed by using the ClonExpress MultiS One Step Cloning Kit (Vazyme Biotech Co., Ltd., Nanjing, China). Plasmid isolation and DNA purification were performed with kits (Axygen Biotechnology Company Limited, Hangzhou, China).

**Construction of plasmids.** The primers and plasmids used in this study are listed in Supplementary Tables 4 and 5, respectively.

The vector expressing the flr gene (A4U99_05915) from F. plautii ATCC 49531 in E. coli DH5α was constructed as follows. The flr gene was PCR-amplified from F. plautii ATCC 49531 genomic DNA using the primers A4U99_05915-s/ A4U99_05915-a. The PCR product was analyzed by agarose gel electrophoresis and the target band was recovered using the DNA gel recovery kit (Axygen Biotechnology Company Limited, Hangzhou, China). The obtained DNA fragment was digested with NheI/BamHI and then inserted into the pET28a vector that was digested with the same restriction enzymes, yielding the plasmid pET28a-A4U99_05915. The constructed plasmid was checked by DNA sequencing. The plasmids for expressing the other genes from F. plautii ATCC 49531 were constructed with the same steps.

The vectors for expressing the mutated flr genes were constructed as follows. Using the plasmid pET28a-FLR-G106A as an example, two DNA fragments, CoFLR-G106A-up and CoFLR-G106A-down, were firstly obtained by PCR-amplification using the primers A4U99_05915-s/CoFR-G106A-a and CoFR-G106A-s/A4U99_05915-a, respectively, and the aforementioned plasmid pET28a-A4U99_05915 as the template. Next, an overlapping PCR using these two DNA fragments, the forward primer A4U99_05915-s and the reverse primer A4U99_05915-a, was carried out with the reagents from the KOD plus Neo DNA polymerase kit (Toyobo, Osaka, Japan), and the generated DNA fragment (CoFLR-G106A) was digested with NheI/BamHI and then ligated with the linear pET28a that was digested with the same restriction enzymes, yielding the target plasmid. The constructed vector was further checked by DNA sequencing.

The vectors for expressing the truncated flr genes were constructed as follows. In brief, PCR-amplification reactions were performed using the aforementioned plasmid pET28a-A4U99_05915 as the template and the corresponding primers with the reagents from KOD plus Neo DNA polymerase kit (Toyobo, Osaka, Japan). The generated DNA fragments were digested with NheI/BamHI and then ligated with the linear pET28a that was digested with the same restriction enzymes, yielding the target plasmids. The constructed plasmids were further checked by DNA sequencing.

**Generation of the *C. ljungdahlii* mutant and its complementation strains**. All the plasmids used in *C. ljungdahlii* DSM 13528 were derived from pMTL83151[44]. The construction of the CRISPR-cas9 editing plasmid pMTLcas-clju_c 30220 for deleting the clju_c 30220 gene (homologous to the *flr* gene of *F. plautii* ATCC 49531) in *C. ljungdahlii* DSM 13528 was performed as follows. Briefly, a linear plasmid pMTLcas-pta that was digested with XhoI/SacI, the two homologous arms that flank the open reading frame of clju_c 30220 (PCR-amplified using the primers 30220-up-s/30220-up-a and 30220-down-s/30220-down-a), and the sgRNA that was designed using online tool (www.benchling.com) to target the clju_c 30220 gene were assembled in one step using the Clonexpress MultiS One Step Cloning kit (Vazyme Biotech Company Limited, Nanjing, China), yielding the target plasmid pMTLcas-clju_c 30220.

The vector expressing the *flr* gene of *F. plautii* ATCC 49531 for functional complementation of the *C. ljungdahlii* Δ*flr* mutant (Δclju_c 30220) was constructed by assembling the following fragments: the $P_{1440}$ promoter that was PCR-amplified from the *C. ljungdahlii* DSM 13528 genomic DNA using the primers P1440-s/P1440-a; the *flr* gene obtained by PCR using the primers CoFLR-s/CoFLR-a and the *F. plautii* ATCC 49531 genomic DNA as the template; and the linear plasmid pMTL83151 generated by KpnI/SmaI digestion. All these DNA fragments were assembled in one step using the Clonexpress MultiS One Step Cloning kit, yielding the pMTL83151-CoFLR plasmid.

The above-mentioned plasmids were transformed into the wile-type *C. ljungdahlii* strain or the Δ*flr* mutant by electroporation. The procedures of electroporation and confirmation of transformants were the same as previously described[43].

**Construction of *E. coli* strains expressing FLR and IFR enzymes**. The above-mentioned pET28a-derived plasmids expressing predicted FLR and IFR enzymes were transformed into the *E. coli* BL21 (DE3) strain by electroporation. The *E. coli* transformants were transferred into liquid LB medium (100 µg mL⁻¹ kanamycin) and cultivated at 37 °C with shaking (200 r.p.m.) until an $OD_{600}$ of 0.8–1.0 was reached. The culture (0.5 mL) was then inoculated to 10 mL anaerobic liquid TB medium[45] and cultivated anaerobically at 37 °C. When *E. coli* cells were grown to an $OD_{600}$ of 0.8, IPTG (1 mM) was added into the medium to trigger the expression of the targeted enzymes. Simultaneously, apigenin was added into the medium with a final concentration of 0.8 mM. *E. coli* cells were further cultivated anaerobically at 16 °C with shaking (100 r.p.m.) for 96 h. Finally, the culture was centrifuged at 10,000*g* for 15 min, and the supernatant was used to assay the concentration of apigenin.

**Protein production and purification**. The plasmids containing the coding sequences for target proteins were transformed into *E. coli* BL21 strain (DE3) for expression. Transformants were grown on agar plates (LB medium containing 50 µg mL⁻¹ of kanamycin). A colony was then inoculated into 2 mL of liquid LB medium (containing 50 µg mL⁻¹ kanamycin) and grown overnight at 37 °C. The overnight culture (2 mL) was inoculated into 200 mL of liquid LB medium (containing 50 µg mL⁻¹ of kanamycin) for further cultivation at 37 °C. When $OD_{600}$ reached ~0.8, the protein expression was induced with 1 mM IPTG at 16 °C. After 16 h, *E. coli* cells were collected by centrifugation at 5, 000*g* for 10 min at 4 °C, and then resuspended in lysis buffer (20 mM Tris-HCl, pH 7.9, 10% glycerol, 500 mM KCl, 10 mM imidazole, and 3 mM DTT). Cells were lysed by using a cell disruptor (French Press, Constant Systems Limited, UK), and the lysate was clarified by centrifugation at 14,000*g* for 1 h at 4 °C. The supernatant was loaded onto a Ni²⁺-Sepharose™ 6 fast flow agarose column (GE Healthcare, Waukesha, WI, USA) for purification. The column was then washed with 30 mL protein solution A (20 mM Tris-HCl, pH 7.9, 10% glycerol, 500 mM KCl, 25 mM imidazole, and 3 mM DTT) followed by a 3 mL protein solution B (20 mM Tris-HCl, pH 7.9, 10% glycerol, 500 mM KCl, 250 mM imidazole, and 3 mM DTT). The elute fractions from solution B (containing target proteins) were applied to a PD-10 Desalting Column containing Sephadex G-25 resin (GE Healthcare, Waukesha, WI, USA) for a rapid washing and removal of small contaminants by using a buffer containing 20 mM Tris–HCl (pH 7.9), 500 mM KCl, 10% (v/v) glycerol, and 3 mM DTT, and were further purified by size-exclusion chromatography using a Superdex 200 column (GE Healthcare, USA) pre-equilibrated with buffer containing 100 mM NaCl, 20 mM Tris-HCl (pH7.9), 3 mM FMN, and 3 mM DTT. The purified protein was stored in 50% (w/v) glycerol at −20 °C in air-tight bottles that were capped and sealed. Peak fractions containing target proteins were collected and concentrated to approximately 10 mg mL⁻¹ for crystallization. The selenomethionine (SeMet)-labeled protein purification was performed as previously described[46].

**Quantification of flavonoids and their degradation products**. The concentrations of various flavonoids and their degradation products were determined as previously described[47], using high-performance liquid chromatography (HPLC) (Agilent 1260 infinity HPLC system) with a ZORBAX SB-C18 column (250 mm × 4.6 mm; particle size, 5 µm) and a UV diode array detector. In brief, all compounds were confirmed by comparing their retention times and UV spectra to those of standard substances. Solution A (methanol and 0.1% formate) and solution B (0.1% formate in water) served as the mobile phases to form a gradient as follows: 5% solution A for 2 min, 5–100% solution A for 10 min, and 100% solution A for

2 min. The flow rate was 1 mL min⁻¹. The concentrations of samples were calculated according to peak area-based calibration curves of various concentrations of standards.

**FLR enzyme assays and kinetics**. All the reagents for enzyme activity assays were deoxidized in advance in an anaerobic chamber. The buffer contained 100 mM Tris-HCl (pH 7.0), 5% DMSO, 8 mM NADH, and 0.8 mM FMN. The 100 µL reaction mixture contained 1 µM FLR, 5 µM NADH-specific FMN oxidoreductase, and 0.1 mM flavones or flavonols, and was incubated anaerobically at 37 °C for 1 h. The reaction was quenched by adding 100 µL methanol, followed by a vortex-mixing step. The samples were centrifuged at 14,000*g* for 10 min, and the supernatant was subject to HPLC analysis to determine the concentrations of flavones and flavonols. Each experiment was repeated in triplicate.

The concentration of the purified FLR protein was determined using the Bio-Rad assay reagent (Bio-Rad Laboratories, Hercules, CA, USA) with bovine serum albumin as a standard. The molecular weight (36.27 kDa) of FLR was calculated using ProtParam from the ExPASy Proteomics Server, and further confirmed by the SDS-PAGE method. Finally, the molar concentration of FLR was calculated by concentration versus molecular weight.

To determine the $K_m$ and $k_{cat}$ values of the FLR enzyme for apigenin, FLR was tested at a concentration ([E]) of 0.06 µM and the substrates (apigenin) were added into the reaction system with various concentrations ([S]). The reaction mixture (100 µL) contained 100 mM Tris-HCl (pH 7.0), 5% (v/v) DMSO, 5 mM NADH, 0.5 mM FMN, 5 µM NADH-specific FMN oxidoreductase (Fre), 0.06 µM purified FLR, and different concentrations of apigenin (2, 4, 6, 10, 20, 40, and 60 µM), and was incubated at 37 °C for 0.5-1.5 h. The reaction was then quenched by adding 100 µL methanol. Each reaction was repeated three times. Initial rates of substrate conversion (*V*) were calculated by linear regression for the data points during the initial 0.5–1.5 h reaction time (the substrate consumption versus time). The observed rate constant ($k_{obs}$) was calculated from $k_{obs} = V/[E]$ and then fit to the standard Michaelis-Menten steady-state equation ($k_{obs} = k_{cat}* [S]/(K_m + [S])$ in Graphpad Prism, thereby yielding $k_{cat}$ and $K_m$ values.

The determination of the $K_m$ and $k_{cat}$ values of FLR for naringenin was the same as that for apigenin, except for a modified reaction mixture that contained 100 mM Tris-HCl (pH 7.0), 5% (v/v) DMSO, 0.5 mM FMN, 0.06 µM purified FLR, and different concentrations of naringenin (2, 4, 6, 10, 20, 40, and 60 µM).

**Crystallization, data collection, and structural determination**. For crystallization screens, SeMet-labeled protein crystals were grown at 20 °C using the setting-drop vapor-diffusion method by mixing 0.4 µL of protein with 0.4 µL of reservoir solution containing 0.1 M magnesium formate dihydrate and 15% (w/v) polyethylene glycol 3350. To obtain the crystals of FLR in complex with substrates, we used SeMet-labeled protein crystals to soak substrates (chrysin, apigenin, and luteolin). Soaking of crystals was performed for 3–24 h with a final concentration of 5 mM. Crystals were directly flash-frozen in liquid nitrogen supplemented with 25% glycerol prior to data collection.

The data of native FLR crystals were collected at Shanghai Synchrotron Radiation Facility (SSRF) beamline BL17U and were processed using HKL3000[48]. SeMet-labeled FLR, FLR-chrysin, FLR-apigenin, and FLR-luteolin crystals were collected at SSRF beamline BL19U. The SeMet-labeled FLR structure was solved by the SAD method. The selenium sites were determined and initial phases were calculated using the HKL3000 package. FMN was then manually built and fitted by Coot[49]. Structures of FLR-chrysin, FLR-apigenin, and FLR-luteolin were solved by molecular replacement using Phenix[50] with the SeMet-labeled FLR structure as the initial model. The chrysin, apigenin, and luteolin molecules in the complexes were built based on their respective 2Fo-Fc and Fo-Fc maps. All structure models were refined with Phenix and manually built using Coot. The model for all structures comprised the entire polypeptide chain excepted for the residues (48–67) that might have been disordered. The data collection and model refinement statistics are shown in Supplementary Table 3.

**Size-exclusion chromatography — multi-angle laser light scattering**. The superdex-200 10/300 column used for the SEC-MALLS analysis was equilibrated with buffer containing 100 mM NaCl, 20 mM Tris-HCl (pH7.9), 3 mM FMN, and 3 mM DTT (filtered through 0.22 µM filter and sonicated), and subsequently the buffer was recirculated through the system overnight at 0.5 mL min⁻¹. Then, 100 µL of FLR solution (~5 mg mL⁻¹) was injected. The signals of absorbance UV280, laser scattering, and differential refractive index were recorded by ASTRA software package version 5.3.2.10 (Wyatt Technologies, USA). The experiments were conducted at room temperature. The absorbance UV280 and differential refractive index curves were processed using the Origin.

**Bioinfomatic prediction of putative *flr* genes in *F. plautii* ATCC 49531**. The reported CHI (GenBank code: KF154734) and PHY (GenBank code: AF548616) protein sequences from the gut bacterium *E. ramulus* DSM 16296 were firstly used as the input for the PSI-BLAST search of the *F. plautii* ATCC 49531 genome. The generated genes encoding homologs to *E. ramulus* CHI and PHY were then input into the STRING web server (https://string-db.org/) to search for their functionally associated genes (neighborhood, fusion, gene co-occurrence) in *F. plautii* ATCC

49531. The obtained genes were considered the potential candidates for *flr*, and then subject to further experimental identification.

**Phylogenetic analysis of FLR homologs**. The tBLASTn search was used to query the amino acid sequences of FLR homologs against the translated nucleotide database. The resulting proteins were aligned using the Clustal W software[51]. The alignment was visualized with MEGA X program[51] and trimmed to generate a final length of 300 amino acids, which generated a maximum-likelihood phylogenetic tree by using the maximum-likelihood approach with the LG model and 300 bootstrap replicates. Finally, the Newick file of phylogenetic tree was uploaded to the Interactive Tree of Life (https://itol.embl.de/) web server tool for further display, manipulations, and annotation.

**Assessment of the abundance of *flr* homologs in human gut microbiome**. The publicly available database, Metaquery (http://metaquery.docpollard.org/)[28], was used to assess the metagenomic abundance of *flr* homologs in the different human populations. The parameters used in Metaquery are as follows: identity >50%, e-value <1e−05, and percent coverage of query and target alignment >70%.

**Construction of sequence similarity network**. The protein sequences of ~2000 predicted or characterized ene-reductases[23], including OYE family of oxidor-eductases (OYE, EC 1.6.99.1), oxygen-sensitive FAD- and [4Fe-4S]-containing enoate reductases (EnoR, EC1.3.1.31), medium-chain dehydrogenases/reductases (MDR, EC 1.3.1.74 and 1.3.1.48), short-chain dehydrogenase/reductase salutar-idine/menthone reductase-like subfamily (SDR, EC 1.1.1.208), and two characterized NADPH-dependent quinone reductases (QnoR), were downloaded from the KEGG database (https://www.kegg.jp/kegg/) by web crawlers using Python. Next, SSNs were generated with the Enzyme Function Initiative (EFI) suite of webtools (http://efi.igb.illinois.edu/efi-est/) using the above protein sequences as the input[52]. Networks were subsequently generated with initial edge values of $10^{-20}$. The generated representative node networks were visualized with Cytoscape 3.5.13[53]. Edge scores were further refined in Cytoscape 3.5.1.

**Quantification of the composition of the simplified microbiota**. qPCR assays were performed to determine the relative abundance of each strain in the simplified gut microbiota as previously described[54]. In brief, all the strains (*C. ljungdahlii* DSM 13528 and its mutants, and the other eight representative gut bacteria), which were stored at −80 °C, were separately transferred to 10 mL of gut microbiota medium (GMM) broth[55] supplemented with 1% mucin, and then incubated at 37 °C for 24 h. When the cells reached exponential phase ($OD_{600} = 0.8$), the same volume (2 mL) of the cultures of these strains were mixed. Then, 20 mL of the mixture was added aseptically to 200 mL of GMM broth containing 1% mucin and 1 mM apigenin. All the incubations were performed anaerobically for 40 h in an anaerobic chamber (Whitley A35 Anaerobic Workstation, Don Whitley Scientific Limited, Bingley, West Yorkshire, UK), with sampling times of 0, 8, 16, 24, 32, and 40 h. Next, 2 mL of each sample was centrifuged at 12,000$g$ for 10 min. Cell pellets were then washed once in 2 mL of physiological solution (NaCl, 8.5 g L$^{-1}$). These cell pellets were used for DNA extraction by using the Magen® HiPure Bacterial DNA Kits (Guangzhou, China) according to the manufacturer's protocol. The obtained genomic DNA was stored at −20 °C. PCR-primer design (including the check of melting temperatures, and the presence of hairpins, self-dimers, and pair-dimers) was carried out by using the SP Designer software[56]. Here, the sequences of *groEL*[57], a major housekeeping gene with abundant sequence data in the Cha-peronin Sequence Database (http://www.cpndb.ca), were used to design species-specific primers. Specificities of primers for strains were confirmed in silico by using Primer-BLAST software[58] and in vitro by PCR analysis. qPCR experiments were carried out by using the Bio-Rad iQ5 real-time PCR detection system (Bio-Rad, Palo Alto, USA). The reaction conditions were as follows: 95 °C for 2 min, followed by 40 cycles of 95 °C for 15 s, 55 °C for 20 s, and 72 °C for 20 s. The cycle threshold (Ct) values were determined automatically by the Bio-Rad iQ5 software. The above experiments, including preparation of cell pellets, DNA extraction, and qPCR assays, were performed in duplicate.

**Q-ToF mass spectrometry measurements**. The 1290 Infinity HPLC system coupled with a 6545 Q-ToF mass spectrometer detector (Agilent, Waldbronn, Germany) was used for analysis. The control software was Mass Hunter Work-station version B.08.00 (Agilent, Santa Clara, CA, USA). The Q-ToF used a Dual Jet Stream Electrospray Ionization (Dual AJS-ESI) source operated in the positive ionization mode and the following parameters were set: capillary voltage, 3500 V; fragmentor, 135; gas temperature, 300 °C; drying gas, 6 L min$^{-1}$; nebulizer, 30 psig; sheath gas temperature, 320 °C; sheath gas flow, 11 L min$^{-1}$; acquisition range, 100–1100 $m/z$; and CID at 0, 10, 20, and 40 V. Samples were analyzed after injection (0.8 μL) on a Zorbax SB C18 column (50 mm × 4.6 mm; particle size, 3.5 μm) and the flow rate was 0.3 mL min$^{-1}$. The solvent system was 0.1% formic acid in water (solvent A) and 0.1% formic acid in methanol (solvent B). The elution gradient (time, % of solvent B) was: 0 min, 5%; 12 min, 95%; 15 min, 95%; and post-time of 2 min.

**UV-Vis spectroscopy**. The FLR protein was diluted to a concentration of 50–100 mM in UV-transparent cuvette. Cary 60 UV-Vis spectrophotometer (Agilent Technologies, Santa Clara, USA) was used to record UV-Vis spectrum of the FLR and FMN standard in the wavelength range from 400 to 800 nm, with a resolution of 5 nm.

**Circular dichroism spectroscopy**. CD experiments were carried out on a Chir-ascan spectrometer (Applied Photophysics Ltd., UK) using 0.16 mg mL$^{-1}$ nar-ingenin product in methanol solution. The ellipticity values (millidegree; mdeg) were recorded at 23 °C between 350 and 210 nm with a 1-nm step size and an integration time of 0.5 s. A total of three accumulations were averaged and the methanol solution spectrum obtained under identical conditions was subtracted. The acquired CD spectra were converted to mean residue ellipticity (mdeg cm$^2$ dmol$^{-1}$) by using the Pro-Data viewer software. The absolute config-uration of naringenin was confirmed based on comparison of its experimental CD with the previously reported data[59,60].

**Reporting summary**. Further information on research design is available in the Nature Research Reporting Summary linked to this article.

## Data availability

The atomic coordinates of FLR-apo, FLR-chrysin, FLR-apigenin, and FLR-luteolin structures have been deposited in the Protein Data Bank with accession codes of 7D39, 7D38, 7D3A, and 7D3B, respectively. The protein sequences of predicted or characterized ene-reductases were downloaded from the KEGG database (https://www.kegg.jp/kegg/). The data on human gut metagenomes were obtained from the MetaQuery (http://metaquery.docpollard.org/) database. Other relevant data supporting the findings of this research are available in the article and the Supplementary Information. In addition, datasets generated and analyzed in the study are available from the corresponding author upon reasonable requests. Source data are provided with this paper.

## Code availability

The amino acid sequences of enzymes (containing EC numbers) for SSN analysis were downloaded from KEGG. The generated source code is available at https://github.com/lovingstudy/kegg_api/blob/master/extract_aaseq_kegg.py (in Python programming language, v3.6.8).

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

## Acknowledgements

This research was supported by the National Natural Science Foundation of China (Nos. 31921006, 31630003), the Shanghai Science and technology Commission (19XD1424500), and Tolo Biotech. Co., Ltd. (Anhui). We thank the staff members at BL17U of Shanghai Synchrotron Radiation Facility and BL19U1 of National Center for Protein Science Shanghai for technical assistance in data collection, the staff at the core facility center of Institute of Plant Physiology and Ecology for X-ray diffraction experiment analysis, and the staff members at National Center for Protein Science Shanghai for their technical assistance on SEC-MALLS data collection and analysis. We also thank Dr. Hua Yuan and Prof. Gongli Tang for helpful discussions on the FLR's catalytic mechanism, Dr. Jing Xing for help with the SSN construction, and Dr. Yining Liu for help on HPLC-MS technical and data analysis.

## Author contributions

G.Y. isolated genes, characterized enzymes, analyzed the phenotypic changes of mutants, and performed bioinformatic and LC-MS analyses. S.H. performed crystal structure analysis of proteins. P.Y. constructed plasmids and mutants. Y.S, Y.W., and P.Z. discussed results and offered advices. W.J. and Y.G. supervised and directed the study. Y.G., W.J., G.Y., S.H., and P.Z. wrote the manuscript.

## Competing interests
The authors declare no competing interests.
