## [Peer Review File · Nature Communications]

REVIEWER COMMENTS

Reviewer #1 (Remarks to the Author):

The manuscript has been much improved, thank you for the detailed response.

Reviewer #4 (Remarks to the Author):

In the manuscript entitled “Discovery of a novel class of ene-reductases for initiating flavonoid catabolism in gut bacteria”, Yang et al describes the discovery of an enzyme capable of catalyzing the first reductions step of flavones and flavonols. Similar to other ene-reductases, it catalyzes the reduction of the conjugated C-C double bond on the heterocyclic C ring. The authors proved this through knockout and complementation studies in vivo, and also in vitro activity. The structure of the flavone reductase (FLR) were solved and kinetic analysis were also performed.

After reading the manuscript and the SI, I had some questions that was answered only by reading the rebuttal letter and comments to the reviewers from the previous round of reviewing. I largely agree with these reviewers, and the authors should have included the additional data in their manuscript or SI, and not only in the point-to-point response (i.e. the growth curves showing the effects of the knockout, the bacterial IFR data, etc). I would strongly urge the authors to re-examine the additional data in the response to reviewers and include this in the SI and briefly referring to it in the main text, as I suspect most readers will have the same questions.

The title might be misleading, as this FLR are only responsible in flavone and flavonol metabolism and not all flavonoids subclasses as given in the introduction.

The low catalytic rate of this enzyme is disconcerting, but also supported in the in vitro experiments, where it took the bacteria more than 24 h to convert 0.1 mM.

Many parts of the figures in the main text can be moved to the SI, as some of it simply does not really add crucial insight. Also, some parts should be revised, especially the close up structures of the active site with the bound substrates in the main as well as the SI. The authors should include the OMIT maps of the co-factors and the substrates. In the SI, the figures should be bigger and distances indicated in Å for crucial interactions. Sometimes, flat projections add to the detail that is sometimes not clear in the 3D structures.

In the proposed mechanism, the authors indicate a specific enantiomer forming. Was the chirality proven?

Minor points:

Please include CC1/2 values of data collection (SI Table)

I do not agree with the term “unorthodox” used in the revision

Page 4, line 79: “in the gut *F. plautii* ATCC 49531”, remove “the gut” or rephrase to “the gut bacteria, *F. pla.*” Other instances of this in the manuscript should also be addressed.

Page 18, line 358, “indicating that FLR and WrbA are far in evolution”. Rephrase as it does not have the intended meaning. Also remember that structure rather than sequence is more indicative of evolutionary relatedness. Pairwise alignment values over the entire protein would probably be less confusing than percentage identity over a small coverage region and lead to less confusion.

Point-by-point response to the referees' comments

Dear Reviewers,

We appreciate your valuable comments on the manuscript (NCOMMS-20-31553A). All the comments and suggestions have been very carefully considered, and the revised points are as listed below.

Reviewer #4 (Remarks to the Author):

In the manuscript entitled “Discovery of a novel class of ene-reductases for initiating flavonoid catabolism in gut bacteria”, Yang et al describes the discovery of an enzyme capable of catalyzing the first reductions step of flavones and flavonols. Similar to other ene-reductases, it catalyzes the reduction of the conjugated C-C double bond on the heterocyclic C ring. The authors proved this through knockout and complementation studies in vivo, and also in vitro activity. The structure of the flavone reductase (FLR) were solved and kinetic analysis were also performed.

After reading the manuscript and the SI, I had some questions that was answered only by reading the rebuttal letter and comments to the reviewers from the previous round of reviewing. *I largely agree with these reviewers, and the authors should have included the additional data in their manuscript or SI, and not only in the point-to-point response (i.e. the growth curves showing the effects of the knockout, the bacterial IFR data, etc). I would strongly urge the authors to re-examine the additional data in the response to reviewers and include this in the SI and briefly referring to it in the main text, as I suspect most readers will have the same questions.*

Reply: Thanks for the suggestion. We have re-checked the content in the response to reviewers and included the additional data in the Supplementary information section, including the growth curve of the *flr*-deleted and wild-type *C. ljungdahlii* strain (Supplementary Fig. 13), the effect of apigenin on the growth of each individual isolate in the microbial community constructed in this study (Supplementary Fig. 14), the activity assay of IFRs (*sgrIFR*, gene ID: SGR_2256) from *Streptomyces griseus* (Supplementary Fig. 1d and 1e), an extra sequence similarity network (SSN) with an increased edge score (Supplementary Fig. 7), and the possible mechanism of apigenin reduction by the FLR enzyme (Supplementary Fig. 15).

The title might be misleading, as this FLR are only responsible in flavone and flavonol metabolism and not all flavonoids subclasses as given in the introduction.

Reply: Thanks for the suggestion. The title has been modified to “Discovery of an ene-reductase

for initiating flavone and flavonol catabolism in gut bacteria”.

The low catalytic rate of this enzyme is disconcerting, but also supported in the in vitro experiments, where it took the bacteria more than 24 h to convert 0.1 mM.

Reply: Thanks for the comments. Actually, this is also an issue of concern in the first round of reviewing. To confirm this result, we tested more FLR-like enzymes and added the data into the last round of revision (Supplementary Figure 6). A possible reason for the low catalytic rate of this enzyme is that the *in vitro* experiment is different from the real intestinal environment and therefore can not show the full potential of this enzyme.

Many parts of the figures in the main text can be moved to the SI, as some of it simply does not really add crucial insight.

Reply: According to the suggestion, Fig. 1b, Fig. 2a, and Fig. 2d have been moved to the SI (Supplementary Fig. 1a, Supplementary Fig. 3, and Supplementary Fig. 4a).

Also, some parts should be revised, especially the close up structures of the active site with the bound substrates in the main as well as the SI. The authors should include the OMIT maps of the co-factors and the substrates.

Reply: Thanks for the comments and suggestions. We have made corresponding changes to Fig. 4c and Supplementary Fig. 10, and added omit maps of the co-factors and substrates (as shown below). The main text has been modified accordingly (Line 242-243).

Fig. 4c, Zoom-in views of the cofactor FMN- and apigenin-binding site. The pictures show the amino acid residues surrounding the active site, and *Fo-Fc* density map(blue), contoured at 1.0 σ level, from which the FMN and apigenin were omitted.

Supplementary Figure 10 | The FLR's structural difference around the substrate-binding pocket in the presence of different substrates. The lower panels are zoom-in views of substrates-binding site. The pictures show F_o-F_c density map (blue), contoured at 1.0σ level, from which the co-factors and substrates were omitted.

In the SI, the figures should be bigger and distances indicated in Å for crucial interactions. Sometimes, flat projections add to the detail that is sometimes not clear in the 3D structures.

Reply: We have revised the corresponding pictures. As shown below, the images are regenerated in PyMOL and the section of ray shadow is turned off. Additionally, the crucial interactions are indicated by corresponding numerical distances in Å.

Supplementary Figure 11 | Crystal structure of FLR dimer and its interface. a, Monomers A and B are shown with ribbon cartoon and electron static surface potential, respectively. The tail is colored cyan. b, c, Zoom in view of amino acid residues interactions at interface. Water is shown as a sphere (red). Distances are shown by dashed lines.

In the proposed mechanism, the authors indicate a specific enantiomer forming. Was the chirality proven?

Reply: According to this suggestion, we have examined the chirality of the naringenin produced from FLR-catalyzed apigenin hydrogenation using circular dichroism (CD) spectrum. As expected, we obtained the CD spectrum of the naringenin product (as shown below, right figure), which is consistent with the previously reported CD spectra of *S*-(-)-naringenin (left figure). This result has been added into the revised manuscript (Supplementary Figure 16).

The reported absorption (UV, thin line) and circular dichroism (CD, thick line) spectra of *S*-(-)-naringenin in ethanol.

Org. Biomol. Chem., 2004, 2, 3602–3607

The UV (black line) and circular dichroism (CD, red line) spectra of the separated *S*-(-)-naringenin in methanol solution.

Minor points:

Please include CC1/2 values of data collection (SI Table)

Reply: The CC1/2 values of data have been added into the Supplementary Table 3.

I do not agree with the term “unorthodox” used in the revision

Reply: Thanks for pointing out this deficiency. The term “unorthodox” has been deleted or changed to “distinct” in the revised manuscript (line 28; line 104; line 167; line 978).

Page 4, line 79: “in the gut F. plautii ATCC 49531”, remove “the gut” or rephrase to “the gut bacteria, F. pla..” Other instances of this in the manuscript should also be addressed.

Reply: Thanks for the suggestion. This place (Page 4, line 79) and other similar instances in the manuscript have been revised.

Page 18, line 358, “indicating that FLR and WrbA are far in evolution”. Rephrase as it does not have the intended meaning. Also remember that structure rather than sequence is more

indicative of evolutionary relatedness. Pairwise alignment values over the entire protein would probably be less confusing than percentage identity over a small coverage region and lead to less confusion.

Reply: Thanks for pointing out this deficiency in expression. To avoid misleading, this sentence has been deleted in revised manuscript (line 368).

REVIEWERS' COMMENTS

Reviewer #4 (Remarks to the Author):

The authors have addressed the issues raised during the review. Some minor corrections still required.

1. The authors should state only Tyr without the residue number for OYE proton donor (Page 20, line 397. Examples of OYE ene-reductases do exist that does not contain the Tyr proton donor, and the proton is from the solvent.
2. The CC1/2 values in SI should be checked, as the value cant be above 1.

Point-by-point response to the referees' comments

Reviewer #4 (Remarks to the Author):

The authors have addressed the issues raised during the review. Some minor corrections still required.

Reply: We appreciate your valuable suggestions on the manuscript. The revised points are as listed below.

1. The authors should state only Tyr without the residue number for OYE proton donor (Page 20, line 397. Examples of OYE ene-reductases do exist that does not contain the Tyr proton donor, and the proton is from the solvent.

Reply: Thanks for the suggestion. We have revised this sentence, in which the residue number of Tyr has been deleted (as shown below).

*“FLR also lacks a direct proton donor to the α -carbon of apigenin for reduction, which is different from OYEs that use **tyrosine residue** as a general acid.”*

2. The CC1/2 values in SI should be checked, as the value can't be above 1.

Reply: Thanks for the suggestion. The CC1/2 values in SI have been revised.

	SeMet-FLR	SeMet-FLR-Chrysin	SeMet-FLR-Apigenin	SeMet-FLR-Luteolin
Data collection				
Wavelength (Å)	0.97850	0.97849	0.97849	0.97849
Space group	P4 ₃ 2 ₁ 2	P4 ₃ 2 ₁ 2	P4 ₃ 2 ₁ 2	P4 ₃ 2 ₁ 2
Cell dimensions				
a, b, c (Å)	67.870, 67.870, 194.270	66.365, 66.365, 194.057	63.969, 63.969, 192.262	65.465, 65.465, 193.544
α , β , γ (°)	90, 90, 90	90, 90, 90	90, 90, 90	90, 90, 90
Number of molecules in ASU	1	1	1	1
Resolution (Å)	30-2.2 (2.28-2.20) ^a	30-2.65 (2.74-2.65)	30-2.55 (2.64-2.55)	30-2.25 (2.33-2.25)
Unique reflections	21222	11627	12318	16462
Redundancy	22.0	21.8	23.1	11.6
I/sigma (I)	34.5(2.33)	30(3.14)	28.67(2.15)	24.4(2)
R _{merge}	0.110(0.522)	0.136(0.701)	0.149(0.870)	0.099(0.970)
Completeness (%)	99.6(96.7)	99.7(97.1)	99.9(99.3)	100(99.8)
CC _{1/2}	0.973	0.990	0.949	0.995